# Using Hybrid HMM/DNN Embedding Extractor Models in Computational Paralinguistic Tasks

**DOI:** 10.3390/s23115208

**Published:** 2023-05-30

**Authors:** Mercedes Vetráb, Gábor Gosztolya

**Affiliations:** 1Institute of Informatics, University of Szeged, H-6720 Szeged, Hungary; ggabor@inf.u-szeged.hu; 2ELKH-SZTE Research Group on Artificial Intelligence, H-6720 Szeged, Hungary

**Keywords:** hidden Markov model, deep neural network, embedding, hybrid acoustic model, computational paralinguistics

## Abstract

The field of computational paralinguistics emerged from automatic speech processing, and it covers a wide range of tasks involving different phenomena present in human speech. It focuses on the non-verbal content of human speech, including tasks such as spoken emotion recognition, conflict intensity estimation and sleepiness detection from speech, showing straightforward application possibilities for remote monitoring with acoustic sensors. The two main technical issues present in computational paralinguistics are (1) handling varying-length utterances with traditional classifiers and (2) training models on relatively small corpora. In this study, we present a method that combines automatic speech recognition and paralinguistic approaches, which is able to handle both of these technical issues. That is, we trained a HMM/DNN hybrid acoustic model on a general ASR corpus, which was then used as a source of embeddings employed as features for several paralinguistic tasks. To convert the local embeddings into utterance-level features, we experimented with five different aggregation methods, namely mean, standard deviation, skewness, kurtosis and the ratio of non-zero activations. Our results show that the proposed feature extraction technique consistently outperforms the widely used x-vector method used as the baseline, independently of the actual paralinguistic task investigated. Furthermore, the aggregation techniques could be combined effectively as well, leading to further improvements depending on the task and the layer of the neural network serving as the source of the local embeddings. Overall, based on our experimental results, the proposed method can be considered as a competitive and resource-efficient approach for a wide range of computational paralinguistic tasks.

## 1. Introduction

Historically, the main research topic of automatic speech processing has been automatic speech recognition (ASR). In ASR, we have to automatically create a transcription for audio (e.g., recording or utterance). From the 1990s to the present, several other topics have received more attention, such as speaker recognition and diarisation (“who’s speaking when”) [1], speech compression [2], cognitive load measurement [3,4], detecting Parkinson’s [5,6,7] or Alzheimer’s [8,9,10] disease, identifying Multiple Sclerosis symptoms [11] or assessing the level of depression [12]. Besides these tasks, a complete subfield has arisen, concerning phenomena present in human speech, containing tasks such as age and gender recognition [13], emotion recognition [14,15], identifying laughter events [16], estimating the degree of sleepiness [17] or conflict intensity [18] and detecting whether the speaker is intoxicated [19]. These subtopics belong to the field of computational paralinguistic, which has recently started to receive more interest. In this field, instead of generating transcriptions, we seek to identify other phenomena present in a speech signal, focusing on the non-verbal content of human speech. It refers to the non-verbal aspects of human communication such as tone of voice and other vocal cues. These cues play an essential role in understanding human communication and can significantly impact the meaning and interpretation of spoken language. Paralinguistic features are often extracted from audio data in machine learning applications, using techniques such as speech analysis and audio signal processing. With the various acoustic sensors becoming increasingly cheaper (and, in parallel, more and more widespread), they could allow the remote monitoring of speaker traits and states. It can opening up a wide range of potential applications such as warning when a vehicle driver is too tired or sleepy. A major boost for paralinguistic was the Interspeech Computational Paralinguistic Challenge (ComParE, later renamed ACM Multimedia Computational Paralinguistic Challenge), which has been held annually since 2009 [20,21,22]. It has led to the development of publicly available datasets and produced a consensus among standard methods, tools and evaluation metrics.

We have to consider slight but really important differences between computational paralinguistic and ASR. One important dissimilarity is the focus of the two area. In automatic speech recognition, we concentrate on the spoken content of the speech signal and try to ignore any other information present (such as the age, gender, native language or the inner feelings of the speaker), as these are considered irrelevant. On the contrary, in computational paralinguistics, we focus on one of the latter speaker states and traits, and disregard the actual words uttered. This dissimilarity of focuses actually leads to a significant technical difference as well.

One technical difference came from the relationship between the length of the input and the output. In ASR the output is the correct phone sequence for a given speech recording and the size of the input utterance is roughly proportional to the length of the output: we expect more words uttered over a longer period of time. Technically, this means that in the traditional speech recognition paradigm the classification step is performed at the local level, handling the audio in small, equal-sized parts called frames. In this case, traditional classifiers such as Gaussian Mixture Models (GMMs [23]), and more recently deep neural networks (DNNs [24]), are used to estimate the local likelihood of the different phones and phone-derived classes from standard frame-level features (such as MFCCs [23]). These local likelihood estimates (the classes of the frames) are then combined over the time axis in the subsequent step (for example, using a hidden Markov model [25]) to obtain the utterance-level output [26] (i.e., a time-aligned sequence of phones). In contrast, in the field of computational paralinguistics, we need to associate different lengths of audio recording inputs with a single label output. Here, the input speech signal is split into larger chunks of continuous speech (such as one sentence), and these chunks are treated as separate units. A given artifact or speaker state (e.g., emotion) is assigned to these chunks (“utterances” or “recordings”). From a machine learning perspective, this means that one such utterance will be one machine learning example. The traditional machine learning models can only process fixed-length inputs, so we need to convert our varying-length recordings into fixed-length feature vectors. We have to calculate a fixed-length feature vector out of these varying-length recording, because, traditional classifiers are unable to handle a concatenation of frame-level attributes of varying lengths as input features. Perhaps the most straightforward solution for that is to take the frames and aggregate the local results instead of combining them. The name of aggregation, is refers to a process rather than a specific mathematical method. The use of this statistic conversion allows us to obtain a fixed-sized utterance-level vector, so the length of the output no longer depends on the length of the input recording.

Another important difference between the two area concerns the size of the databases. For ASR nowadays, hundreds or even thousands of hours’ worth of databases are available, meaning tens or hundreds of millions training examples for frame-level phoneme classification. This allows researchers to train DNN models, which are known to be data-greedy. By comparison, computational paralinguistics typically has small corpora, because each task usually requires specific recording protocols and annotations. This drawback means that usually there are just a few hundred (or at most a few thousand) examples for a specific paralinguistic subtopic or class. Therefore, traditional classifier models (which can be well trained on very few data) are used instead of end-to-end neural networks. It is common to employ learning methods for classification such as Support Vector Machines (SVM [27]). Deep neural network machine learning techniques (e.g., fine-tuning) are only rarely used in computational paralinguistics. Standard solutions are still dominant, for example low-level descriptors (e.g., energy, spectral and cepstral (MFCC)) and voicing-related attributes for frame-wise computing; statistic conversion techniques, such as mean and standard deviation for aggregation; and classification methods such as SVMs. In the last decade, there have been few research studies that have applied complex machine learning models and usually the performance is strongly task-dependent.

Based on our previous studies [28,29], we developed a method shown in Figure 1 that combines ASR and paralinguistic approaches. For frame-wise computing, we followed standard ASR principles and we used DNNs to perform a frame-level feature extraction. Afterwards, to aggregate these features, we used more or less traditional computational paralinguistics techniques such as standard deviation and kurtosis. In the end, we employed SVM models to perform the classification task.

## 2. Processing Paralinguistic Data

Figure 1 shows the complete workflow of our experimental method from the preprocessing of the ASR corpus to the classification of the paralinguistics corpus. As mentioned above, we created a hybrid method that follows ASR principles and computational paralinguistics principles too. In this section, we will describe the workflow in a step-by-step fashion.

### 2.1. HMM/DNN Hybrid Model

Hidden Markov models (more specifically, HMM/GMMs) used to be the state of the art in automatic speech recognition. They consisted of a local GMM module, being responsible for supplying local (i.e., frame-level) phonetic probability estimates, while the HMM part was responsible for combining these local estimates into utterance-level phone sequences [30]. After deep neural networks were invented, these HMM/GMM models were developed into HMM/DNN [31] hybrid models by replacing the GMM component with a deep neural network, still operating locally (i.e., on the frame level). Soon it became widespread knowledge how to efficiently train and employ HMM/DNN hybrid models. In our study, we seek to employ this knowledge by training such a DNN acoustic model, and using this as the base of our feature extractor for computational paralinguistic tasks.

The HMM/DNN model has two parts. The first part is the deep neural network, while the second part is a hidden Markov model. The outputs of the DNN will be the input of the HMM. The DNN gives frame-level estimations, which will be a posterior probability ($P(c_{k}|x_{i})$). The next part, the HMM, expects a class-condition likelihood ($p(x_{i}|c_{k})$), so before utilising the output of the DNN in the HMM, we have to transform it. The transformation can be processed with Bayes’ theorem. If the posterior estimation is divided by a priori probabilities of phonetic classes ($P(c_{k})$); then, we obtain class-condition likelihood value within a scale factor. The a priori probabilities are usually estimated using simple statistical methods. However, the scale factor can be ignored because it has no influence on the subsequent search process.

Nowadays, recurrent neural architectures have become the state of the art in ASR [26]. Applying units such as long short-term memory (LSTM) [32] and Gated Recurrent Units (GRU) [33] as building blocks leads to a better performance. Nevertheless, there are several reasons for employing an HMM/DNN model instead of applying a recurrent neural network. The simple feed-forward DNN structure employed in the HMM/DNN acoustic model makes the training steps easier: it has lower computational complexity and uses less memory. These networks still have a competitive performance [34,35] in the case where training data are scarce.

### 2.2. DNN Embedding Extraction

To extract embeddings for further classification, first we have to train our hybrid model. We can see the acoustic HMM/DNN model training in the left top corner of Figure 1. Here, we need a larger ASR corpus that has time-aligned phonetic labels. From this corpus, we have to extract frame-level features. The extraction can be handled using different techniques, such as calculating filter banks, deltas, spectrograms or using neural networks. Now, we can use these frame-level features to train our hybrid model for a general language structure. When the training phase is over, we need to make a slight modification to our model to use it for DNN embedding extraction. We have to detach the DNN from the hybrid model and fix its weights. In this case, we are not interested in the original output layer of our DNN, which produces the posterior estimates. Now, we will focus on the previous hidden layers and their activation values, because hidden layers can provide more abstract information. We can see the process of embedding extraction in the left bottom corner of Figure 1. Here, we have to extract frame-level features from a smaller paralinguistics corpus. The length of a frame-level feature has to be the same as a feature from the ASR corpus. The best way to achieve this is to use the same method here as before. Afterwards, we can feed them into the modified deep neural network. The output of the hidden layers will be our embedding features.

### 2.3. DNN Embedding Aggregation

When we have acoustic frame-level embeddings, we have to convert them into utterance-level features in order to perform a classification. Figure 1 shows the final classification workflow in the bottom right corner. Since databases contain recordings with different lengths, we have a different number of embedded features for each recording. Traditional classifiers handle only fixed-sized inputs for one utterance, so we cannot create utterance-level features with a simple frame-level concatenation. We need to aggregate embeddings and Figure 2 shows the method in more details. This could be performed in a straightforward way by calculating statistical values along their time axis, such as mean, standard deviation or others. The final size of the aggregated vector is independent of the length of the original recording and it only depends on the number of neurons in the last given hidden layer and on the aggregation technique used. In the end, these utterance-level feature vectors can be fed into any traditional classification or regression model, where the output will be a label (class or real number) for each recording.

## 3. The Databases Used

Next, we will introduce the datasets that we employed in our experiments. Different aspects were taken into account when selecting the databases. On the one hand, we preferred databases that were easily accessible to the research community, and thus the databases used in the ComPare challenge were chosen. The other aspect was that they should be easily comparable, which is why we chose three German language databases so possible language differences would not affect the results. To cover different topics, we used three paralinguistic corpora (AIBO, URTIC and iHEARu-EAT). Although these corpora cover different topics, the recording conditions (such as sampling rate, language and background noise) are quite similar. Table 1 showes a summary from these three paralinguistic database. The fourth database utilized in our experiments (called BEA) was not a paralinguistic one, but it was used for training our hybrid acoustic model.

### 3.1. AIBO

The FAU AIBO Emotion Corpus [36] contains speech taken from 51 native German children. The children were selected from two schools. The database contains 9959 recordings from the Ohm school, and 8257 recordings from the Mont school. The total duration is approximately 9 h. The subjects had to play with a pet robot called AIBO. They were told that AIBO responds to their commands, but it was actually remotely controlled by a human. The Ohm school recordings are commonly used for training (with speaker-wise cross validation). The Mont school recordings were used for the test set. Because of the size of the training set, we were able to define a development set. We kept recordings of 20 children in the training set (7578 utterances) and used recordings of 6 children in the development set (2381 utterances). The original 11 emotional classes were merged to form a 5-class problem. The new classes were constructed from the originals: Anger (angry, irritated, reprimanding), Emphatic, Neutral, Positive (motherese and joyful), and the Rest (helpless, surprised, bored, non-neutral but not belonging to the other categories). This database was also employed in the INTERSPEECH 2009 Emotion Challenge.

### 3.2. URTIC

The Upper Respiratory Tract Infection Corpus (URTIC) [37] was provided by the Institute of Safety Technology, University of Wuppertal in Germany. It contains native German speech from 630 subjects (248 female, 382 male). The total duration is approximately 45 h. The recordings have a sampling rate of 44.1 kHz downsampled to 16 kHz. They were split into 28,652 chunks of 3 to 10 s. The participants had to complete different tasks. They had to read short stories (e.g., a well-known story in the field of phonetics “The North Wind and the Sun”), had to produce voice commands (such as stating numbers from 1 to 40) and they also had to narrate spontaneous speech (e.g., say something about their best vacation). The number of tasks varied for each speaker. The database was split speaker-independently into training, development and test sets where each one contained 210 speakers. The training and development sets contained 37 infected participants and 173 participants with no cold. There were two classes, namely cold and no cold. The purpose of the classification was to decide whether the speaker had a cold. This database was also employed in the INTERSPEECH 2017 Computational Paralinguistics Challenge.

### 3.3. iHEARu-EAT

The iHEARu-EAT corpus [38] was provided by the Munich University of Technology. It contains close-to-native German speech taken from 30 subjects (15 female, 15 male). It was recorded in a quiet, slightly echoing office room. It contains approximately 2.9 h of speech (sampled at 16 kHz). The recordings were segmented into roughly equal parts. The participants had to perform speaking exercises while eating different type of foods. Speakers had to complete different tasks, e.g., read the German version of “The North Wind and the Sun” story, and they had to give a spontaneous narrative about their favourite activity or place. The number of completed activities varied for each speaker because not everyone was willing to eat every type of food offered. The database was split speaker-independently into a training set (20 speakers) and test set (10 speakers). There were seven classes determined by the consistency: apple, nectarine, banana, crisp, biscuit, gummy bear and without any food. The aim of the classification was to recognise what the subject was eating while speaking. These type of foods typically allowed the participants to eat while speaking. This database was also employed in the INTERSPEECH 2015 Computational Paralinguistics Challenge.

### 3.4. BEA

We used a subset of the BEA Hungarian corpus [39] to pretrain our acoustic model. This was not a specific paralinguistics corpus like the three above, but it is also a speech corpus. It contains only spontaneous speech and it is good for generalising a neural network for speech processing. We applied a subset of this database, which contained the speech of 165 subjects (≈60 h). This subset contained only spontaneous speech with special events such as filled pauses, breathing sounds, laughter, gasps and so on. It had a transcription, where the phonetics set and the special events were also marked.

## 4. Experimental Setup

### 4.1. Frame-Level Features

For both the ASR and paralinguistics corpora, the frame-level feature extraction was carried out by 40 Mel-frequency filter banks with the standard values of 25 ms window width and 10 ms frame step. We also extended it with the log-energy value, and calculated the first- and second-order derivatives (i.e., Δ and ΔΔ [40]). The final number of features in a frame-level vector was 123.

### 4.2. HMM/DNN Hybrid Model

We trained the hybrid model with the large BEA corpus. It has a standard feed-forward deep neural network (DNN). During DNN training and evaluation, we used the standard solution of applying a 15-frames-wide sliding window, so the input layer of the network contained 15×123=1845 neurons. The DNN has five hidden layers, where each one contains 1024 ReLU neurons. The final softmax layer of the network had as many neurons as the number of phonetic states, namely 911. For embedding extraction, we used the activation values of the middle five hidden layers (i.e., layer 1, 2, 3, 4 and 5). Each layer generated 1024-sized frame-level feature vector.

### 4.3. Embedding Aggregation

In the conversion step, we transformed the frame-level embeddings into utterance-level feature vectors by aggregating them with a statistical function along the time axis. The statistical approaches used were the following: arithmetic mean, standard deviation, kurtosis, skewness and “zero ratio”. Zero ratio represents how many times out of each embedding an output neuron fired (it means a feature has a non-zero value, as we used ReLU neurons). The used aggregations has the following mathematical formula, if we have *N* frame-level embeddings in the form $x_{1},x_{2},\ldots ,x_{i},\ldots ,x_{N}$:Arithmetic mean: $\bar{x}=\frac{1}{N}\sum_{i=1}^{n}x_{i}$Standard deviation: $\sigma =\sqrt{\frac{1}{N}\sum_{i=1}^{N}{\left(x_{i}-\bar{x}\right)}^{2}}$Kurtosis $=\frac{1}{N}\sum_{i=1}^{N}\frac{{\left(x_{i}-\bar{x}\right)}^{4}}{\sigma^{4}}$Skewness $=\frac{1}{N}\sum_{i=1}^{N}\frac{{\left(x_{i}-\bar{x}\right)}^{3}}{\sigma^{3}}$Zero ratio $=\frac{1}{N}\sum_{i=1}^{N}y_{i}$, where $y_{i}=\left\{\begin{array}{cc} 1 & \mathrm{if}\;x_{i}>0,\\ 0 & otherwise \end{array}\right.$

Notice that, having *N* embeddings with *m* frames, any of these formulas produce *m* utterance-level aggregated features.

### 4.4. Classification

For optimal results, we separated all paralinguistic corpora into training, development and test sets. We determined the optimal parameters of the classifier while training with the training set end evaluating with the development set. After the optimisation, we measured the overall efficiency while training with the combination of training and development sets and evaluating with the test set. During the classification step, our classifier was a Support Vector Machine (i.e., SVM) [41]. We optimized the complexity parameter using 10 powers between $10^{-5}$ and $10^{0}$. In the case of AIBO and URTIC, we always standardised and downsampled the actual training set before feeding it into the SVM. In the case of iHEARu-EAT, we only performed a speaker-wise standardization.

To measure the efficiency, we calculated the Unweighted Average Recall (i.e., UAR) [42] from the posteriors of our SVM model. UAR measures the average recall across all classes without considering class imbalance. To calculate UAR, you compute the recall for each class and then take the average across all classes. Recall, also known as sensitivity or true-positive rate, is the proportion of true-positive instances (correctly identified instances) out of all actual positive instances. UAR is called “unweighted” because it treats each class equally, regardless of class size or prevalence. This makes it suitable for datasets with imbalanced class distributions, where some classes may have significantly fewer instances than others. It provides a balanced view of the overall performance of a classification system, taking into account the performance across all classes equally. In an emotion recognition task with an imbalanced dataset (Happy: 500, Sad: 300, Neutral: 2000), accuracy can be misleading. For instance, a classifier that predicts the majority class (Neutral) for all instances would have high accuracy (2000/2800≈71.4%). However, UAR (Unweighted Average Recall) gives a better evaluation by considering the recall for each class separately. In this case, UAR would indicate poor performance (UAR: (0+0+1)/3≈0.333) as the classifier fails to identify instances of the minority classes (Happy and Sad) while performing well on the majority class.

### 4.5. Baseline Method

X-vector networks [43] nowadays are receiving more attention in the field of paralinguistics. Previous studies have successfully applied x-vector embeddings in various paralinguistic tasks [5,17]. This feed-forward deep neural network was originally designed for speaker identification, but with a slight modification it can be used for feature extraction as well [43]. Figure 3 shows the structure of our baseline network. It has nine layers, in the following order: five time-delayed frame-level layers, one statistics pooling layer, two segmentation layers and one softmax layer. The statistics pooling layer is responsible for transforming frame-level information into utterance-level information. It aggregates over the output of the fifth frame-level layer and calculates different metrics such as mean or standard deviation. The segmentation layers can capture meaningful information about the speaker, e.g., age and gender. The last layer of the network contains the speaker id.

In order to use x-vector DNNs for paralinguistic feature extraction, we trained the model on the BEA corpus and we evaluated separate baselines for each of the AIBO, URTIC and iHEARu-EAT databases. In parallel, we calculated different frame-level features from each database, such as mfcc, fbank and spectrogram values. After training different x-vector models on them, we fixed the weights of the models and removed the last two layers from each. We extracted utterance-level information from the seventh hidden layer as a network embedding and examine traditional classification on them. We also tried out a noise augmentation technique. To find the best baseline, we always carried out a quick search of the frame-level feature sets with and without augmentation for each database separately.

## 5. Experimental Results

### 5.1. AIBO

When we used the AIBO database, we had a five-class classification task. Figure 4 shows all the results with different statistic conversion techniques on the development set. The best baseline result was obtained using f-bank features with augmentation and it came to 39.3% (indicated by the grey horizontal line in the figure).

Regarding the layers, we can state that the fourth layer always outperforms the baseline. Moreover, the fourth layer achieved the best performance scores with all the aggregation techniques used. In view of aggregation, there were no significant differences between the robustness of aggregations. Kurtosis and skewness had the worst performance scores overall. In the majority of cases, we cannot beat the baseline with them. The mean and standard deviation performed the best. In most cases, they outperformed the x-vector baseline. The average performance of a conversion technique is represented by a black column. The mean and standard deviation of the fourth and fifth layers had better performance scores than their average layer performance scores and these gave the best results overall.

**Figure 4 sensors-23-05208-f004:**
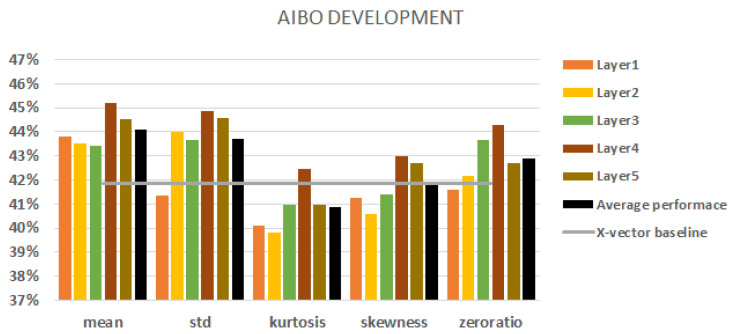
AIBO results on the development set. Extracting embeddings from each of the 5 layers. The black columns denote the average performance of all five layers. The baseline is represented by a grey line.

### 5.2. URTIC

When we used the URTIC database, we had a two-class classification task. Figure 5 shows all the results obtained with different aggregations with the development set. The best baseline result was obtained using MFCC features with augmentation and it was 66.9% (indicated by the grey horizontal line in the figure).

Here we can state that the third and fourth layers always reach or outperform the average performance of a conversion technique (represented by a black column), but in the majority of cases they cannot beat the x-vector baseline. The standard deviation is a bit more robust than the others, but again there is no significant difference. The kurtosis and skewness statistic conversions again had the worst performance scores. Here, the best results can beat the baseline. One of them is the mean of the third and fourth layers. The other is the zero ratio statistic conversion for the second and fourth layers.

**Figure 5 sensors-23-05208-f005:**
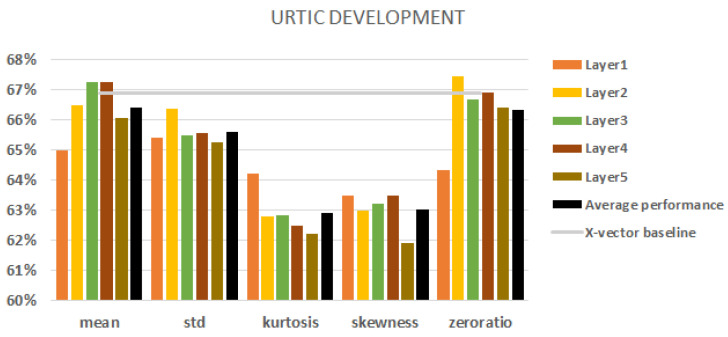
URTIC results on the development set. Embeddings from each of the 5 layers. The black columns denote the average performance of all five layers. Baseline represented by a grey line.

### 5.3. iHEARu-EAT

With the iHEARu-EAT corpus, we had a seven-class classification task. Figure 6 shows all the results obtained with different aggregations using the development set. The best baseline result was obtained using fbank features with augmentation and it was 58.7% (indicated by the grey horizontal line in the figure).

Here, we can state that all of our embeddings always outperform the baseline. Similar to URTIC, the second and fourth layers perform best and in most cases outperform the local average performance (represented by a black column). The robustness behaviour is similar, but the zero ratio and mean are slightly better. The rest of the aggregations behave just like before. The mean and the standard deviation of the second and fourth layers give the overall best results.

**Figure 6 sensors-23-05208-f006:**
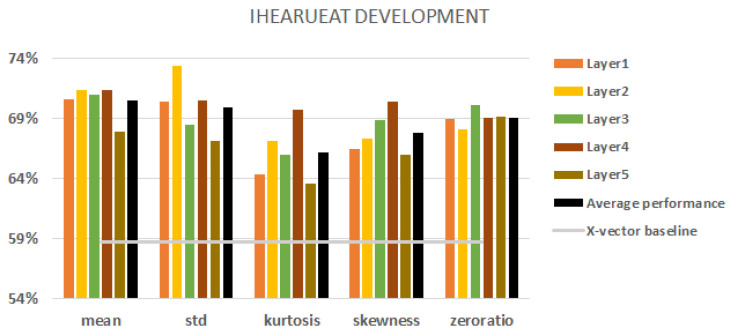
iHEARu-EAT development set results. Embeddings from each of the 5 layers. The black columns denote the average performance of all five layers. Baseline represented by a grey line.

## 6. Combined Results

A summary of our results from the first series is represented in Table 2. We can see that the HMM/DNN embeddings always outperform the x-vectors. The kurtosis and skewness aggregations perform the worst. The mean, standard deviation and zero ratio techniques behave slightly the same. We also wanted to know the expressive power of the embeddings, so we combined all of the five techniques as well. With this, we improved performance on the development set, but the scores of the test sets dropped. This raises the question of whether there is a specific combination of statistic conversation techniques that gives improvements on the development set while maintaining the ability of generalisation. We tried to determine the best-generalised model that would give better performance scores against future data.

In the second series of experiments, we used sequential forward selection (SFS) to combine multiple aggregated feature vectors. The basic idea behind SFS is to initialise the subset with just the best method, and then iteratively add one more aggregation to the subset based on which combination provides the greatest improvement in performance. To combine a subset of aggregations, we concatenated their utterance-level feature sets. The size of each utterance-level feature vector was as follows: 1024 as one technique, 2048 as a concatenation of two different aggregated vectors, 3072 as a concatenation of three different aggregated vectors, 4096 as a concatenation of four different aggregated vectors and 5120 when we concatenated all the different aggregated vectors.

### 6.1. AIBO

Figure 7 shows the performance scores obtained when we combined the mean statistic conversion technique with all the others. The first chart shows the mean aggregation with layer 4. All of the combinations perform better than their x-vector baseline; however, the mean technique had the best performance scores of 45.2% on the dev set and 44.0% on the test set. The second chart shows the mean statistic conversion technique with layer 5. All of the combinations performed better here as well. The combination of mean, skewness, standard deviation and kurtosis gave the best performance score of 45.3% on the dev set and 44.2% on the test set, but we can obtain almost the same without the kurtosis of 45.1% on the dev set and 43.7% on the test set. We can state that the fifth layer can generalise better if we use the combination of mean+ skewness+ standard deviation+kurtosis techniques, and the fourth layer performs best with only mean statistic conversion. The mean and standard deviation techniques always gave improvements. Instead of the fact that layer 5 had better performance scores than layer 4, we should note that calculating just one aggregation requires less time and memory.

### 6.2. URTIC

Figure 8 shows the performance scores obtained when we combined the zero ratio statistic conversion technique with all the others. We can see on the first chart the zero ratio aggregation with layer 2. The best combination (zero ratio+ mean+ standard deviation) had the same performance score (67.4%) on the dev set as the zero-ratio-only option. Note that they have the same performance with the development set, but the combination gives a better performance score (69.6%) on the test set. We can see on the second chart the zero ratio aggregation with layer 4. The first three combinations can outperform the x-vector baseline. If we combine two techniques, the best combination (zero ratio + mean) gives 67.7% with the dev set and 69.5 with the test set. We can state that the fourth layer has the best generalisation if we use the combination of mean and zero ratio techniques. Kurtosis and skewness aggregations always underperform the others. The standard deviation metric can improve the performance with layer 2.

### 6.3. iHEARu-EAT

Figure 9 shows the performance scores when we combine the standard deviation (i.e., std) statistic conversion technique with all the others. The first chart shows the std aggregation with layer 2. All of the combinations perform better than their x-vector baseline on the development and test sets. In the case of combining four techniques, the zero ratio slightly improves our model and increases its ability to generalise. The best combination is std+kurtosis+zero ratio+mean and it produced a 74.9% performance score on the dev set and 78.3% on the test. The second chart shows the std aggregation with layer 4. All of the combinations perform better here as well. When we combined three techniques (std+skewness+zero ratio), it slightly improved our model and gave a 76.0% performance score on the dev set and 75.0% on the test set. We can state that model trained on features from the second layer can generalise better if we use the combination of mean, zero ratio and skewness techniques. The zero ratio always produces a good improvement.

A summary of our results from the second series is given in Table 3. We can say that extracting embeddings from the fourth layer always gives the best performance scores. Concatenating aggregations is always a good idea and it helps our model to generalise better, but we should carefully select the techniques used because the best combination may be task-dependent. We should always consider including the mean, standard deviation and/or zero ratio in the combination.

## 7. Discussion

We trained a state-of-the-art hybrid acoustic HMM/DNN model on a large ASR corpus and then used the DNN part to extract frame-level embeddings from smaller paralinguistics corpora. Afterwards, to aggregate these features into utterance levels, we used statistics computational techniques. We chose traditional aggregations, such as standard-deviation and mean, and less traditional ones, such as skewness, kurtosis and zero ratio; these aggregated vectors served as features in our computational paralinguistic classification experiments. In our first experiments, we tested all aggregation techniques individually. Our results indicate that the hybrid acoustic model performed better than the x-vector did. The mean, standard deviation and zero ratio techniques achieve practically the same performance scores.

After obtaining these results, we wanted to improve the expressive power of the embeddings. We chose to investigate the performance of combined aggregation techniques. We tested all the possible combinations of the five techniques. Our results indicate that we were successfully able to extract features from different paralinguistic tasks with our HMM/DNN hybrid acoustic-model-based feature extraction method. Using the second or the fourth layer of the model is always a good choice. As for aggregations, the mean, standard deviation and zero ratio always help improve the performance, but we have to combine these techniques carefully. In the case of kurtosis and skewness aggregations, we can observe varied behaviour in all databases. In the first stage of our research, when we tested each aggregation separately, we could see the trends amongst them. They had the worst performance in terms of each database and each layer. In the second stage of our research, when we tested the combination of aggregations, we could see a similar tendency. When we were deciding which aggregation should be combined next, skewness or kurtosis gave the lowest scores in 13 of 18 cases. Based on these results, it can be stated that skewness and kurtosis aggregation techniques are not able to significantly improve the success rate of paralinguistic task processing.

In the case of aggregation combination, we can see that combining three techniques will always improve our results in any paralinguistic task. On the other hand, the combination of four techniques will behave inconsistently. Although it improves the results on the development set, the results on the test set are often decreasing. This suggests that the generalisation ability of our model is also decreasing. For this reason, when choosing the number of aggregations to combine, it is worth taking into account Occam’s razor principle, which states that unnecessarily complex models should not be preferred against simpler ones.

The possible limitations of our approach include the potential language dependency of the extracted embeddings. For further research directions, we see several opportunities. Although our results were competitive even with a Hungarian HMM/DNN hybrid acoustic model for German tasks, and although the x-vector method (used as feature extractor) showed language-independency tendencies before [17], the effect of using an acoustic model trained on the same language should be studied in the future. On the other hand, each of the databases studied here is German-speaking, it could potentially be investigated whether aggregations computed from x-vector embeddings behave similarly on databases of different languages. Additionally, it is unclear how the amount of training material affects the quality of the extracted features. Furthermore, training a DNN is inherently a stochastic procedure due to random weight initialization; therefore, the variance in the classification performance might also prove to be an issue. We plan to investigate these factors in the near future. Another possible direction is whether these aggregations computed from different neural network embeddings follow a similar trend.

## Figures and Tables

**Figure 1 sensors-23-05208-f001:**
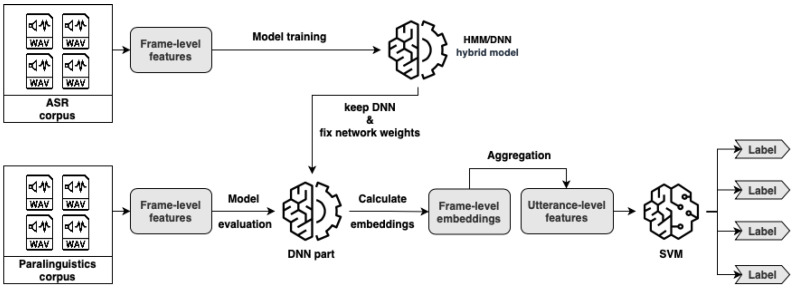
Hybrid HMM/DNN model workflow for paraliguistics task.

**Figure 2 sensors-23-05208-f002:**
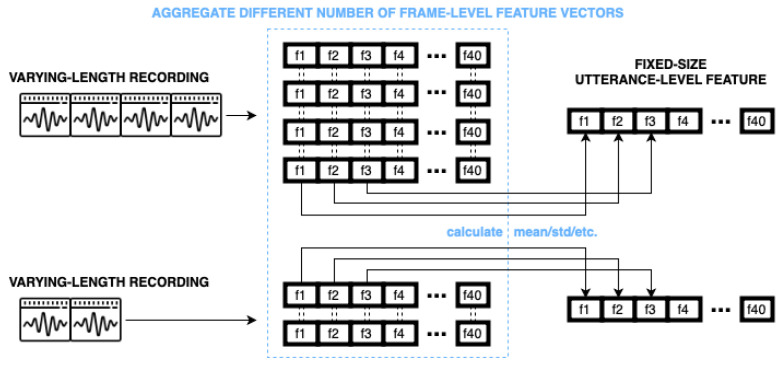
Creating fixed sized feature vectors with statistic conversion.

**Figure 3 sensors-23-05208-f003:**
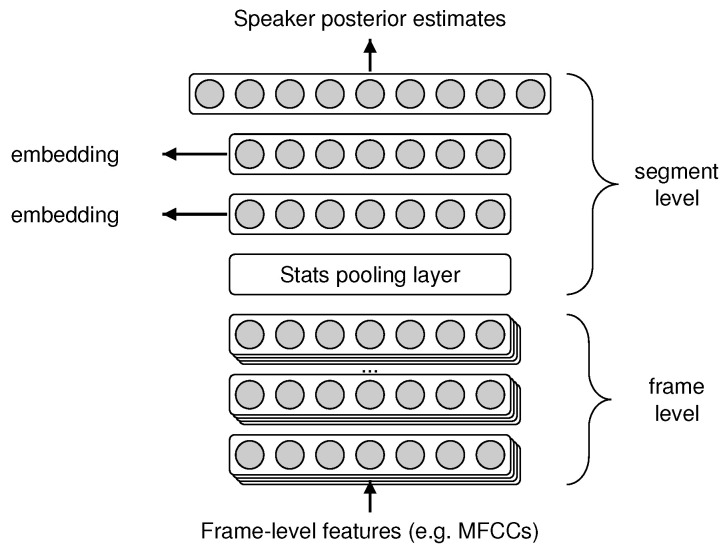
The x-vector neural network structure used as the baseline.

**Figure 7 sensors-23-05208-f007:**
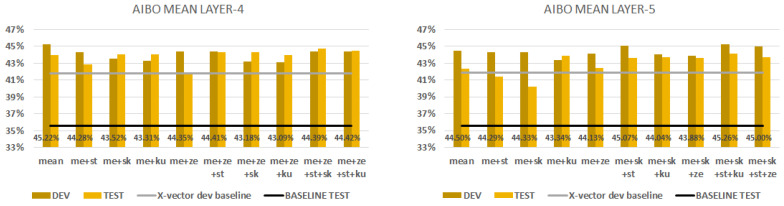
AIBO database classification results with development and test sets. We combine the mean aggregation by SFS. We extract embeddings from layer 4 (first figure) and from layer 5 (second figure).

**Figure 8 sensors-23-05208-f008:**
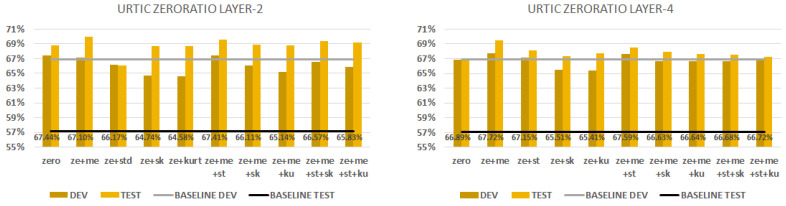
URTIC database classification results with development and test sets. We combine the zero ratio aggregation by SFS. We extract embeddings from layer 2 (first figure) and layer 4 (second figure).

**Figure 9 sensors-23-05208-f009:**
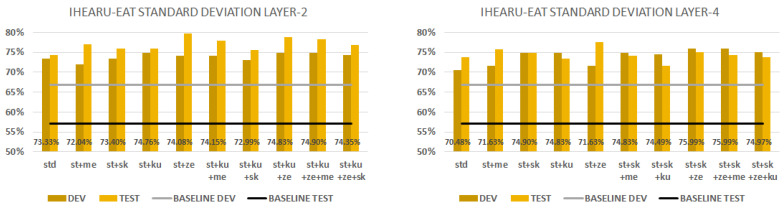
iHEARu-EAT database classification results with development and test sets. We combine the standard deviation aggregation by SFS. We extract embeddings from layer 2 (first figure) and layer 4 (second figure).

**Table 1 sensors-23-05208-t001:** The number of speakers and utterances for the three paralinguistics databases used.

Dataset	Language	No. of Classes	No. of Utterances			No. of Speakers			Total Duration (hh:mm:ss)		
			Train	Dev	Test	Train	Dev	Test	Train	Dev	Test
AIBO	German	5	7578	2381	8257	26	6	25	03:45:18	01:11:46	03:53:44
URTIC	German	2	9505	9596	9551	210	210	210	14:41:34	14:54:47	14:48:18
iHEARu-EAT	German	7	657	287	469	14	6	10	01:20:37	00:31:44	01:00:39

**Table 2 sensors-23-05208-t002:** Results of different aggregation techniques with the three different corpora.

	AIBO			URTIC			iHEARu-EAT		
	Layer	DEV	TEST	Layer	DEV	TEST	Layer	DEV	TEST
mean	4	45.2%	44.0%	4	67.3%	69.3%	2	71.4%	79.0%
std	4	44.8%	44.4%	2	66.4%	68.1%	2	73.3%	74.4%
kurtosis	4	42.5%	40.3%	1	64.2%	60.8%	4	69.7%	69.0%
skewness	4	43.0%	41.2%	1	63.5%	68.3%	4	70.3%	67.3%
zero ratio	4	44.3%	42.1%	2	67.4%	68.8%	3	70.0%	75.5%
all	5	45.5%	44.2%	4	66.0%	65.3%	4	76.6%	74.6%
x-vector	–	41.8%	35.6%	–	66.9%	57.1%	–	58.7%	53.8%

**Table 3 sensors-23-05208-t003:** The best results obtained by SFS. The base aggregation and layers came from the best corpus-specific aggregations.

AIBO				URTIC				iHEARu-EAT			
**Layer**	**Combination**	**DEV**	**TEST**	**Layer**	**Combination**	**DEV**	**TEST**	**Layer**	**Combination**	**DEV**	**TEST**
4	mean	45.2%	44.0%	2	zero	67.4%	68.8%	2	std	73.3%	74.4%
4	me-ze	44.4%	42.0%	2	ze-me	67.1%	70.0%	2	st-ku	74.8%	75.9%
4	me-ze-st	44.4%	44.3%	2	ze-me-st	67.4%	69.6%	2	st-ku-ze	74.8%	78.9%
4	me-ze-st-ku	44.4%	44.5%	2	ze-me-st-sk	66.6%	69.4%	2	st-ku-ze-me	74.9%	78.3%
5	mean	44.5%	42.3%	4	zero	66.9%	67.1%	4	std	70.5%	73.8%
5	me-sk	44.3%	40.2%	4	ze-me	67.7%	69.5%	4	st-sk	74.9%	74.8%
5	me-sk-st	45.1%	43.7%	4	ze-me-st	67.6%	68.5%	4	st-sk-ze	76.0%	75.0%
5	me-sk-st-ku	45.3%	44.2%	4	ze-me-st-sk	66.8%	67.9%	4	st-sk-ze-me	76.0%	74.3%

## Data Availability

No new data were created or analyzed in this study. Data sharing is not applicable to this article.

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
