# Peer review of "Using Hybrid HMM/DNN Embedding Extractor Models in Computational Paralinguistic Tasks"

_sensors, 2023, doi:10.3390/s23115208_

Round 1

Reviewer 1 Report

As per the paper, your proposed extraction technique consistently outperforms the x-vector method. Besides that, you have indicated that your method carries a number of additional advantages compared to the existing ones. You have used three paralinguistic corpora (AIBO, URTIC and iHEARu-EAT). 

Please explain the following statement a bit in detail 

The kurtosis and skewness statistic conversions behave differently for 383

each database, but their performance scores were always poor. If we increase the size of 384

our feature vector with a combination, we can train better classifiers, but beyond a certain 385

point using more dimensions will start to reduce the ability of generalisation. Therefore, 386

when we choose a combination, we should keep in mind the principle of Occam’s razor. 

Author Response

Dear Reviewer,

First, we would like to thank you for your review. 

We would like to clarify the following sentence. We will rewrite this section in our paper too. 

"The kurtosis and skewness statistic conversions behave differently for 383
each database, but their performance scores were always poor. If we increase the size of 384
our feature vector with a combination, we can train better classifiers, but beyond a certain 385
point using more dimensions will start to reduce the ability of generalisation. Therefore, 386
when we choose a combination, we should keep in mind the principle of Occam’s razor."

In case of kurtosis and skewness aggregations we can observe varied behaviour in case of all database. In the first stage of our research, when we tested each aggregation separately we can see the tendency about them. They has the worst performance in case of each database and each layer. In the second stage of our research, when we tested the combination of aggregations, we can see a similar tendency. When we decided about which aggregation should be combined next, skewness or kurtosis gives the lowest scores in 13 of 18 cases. Based on these results, it can be stated that skewness and kurtosis aggregation techniques are not able to significantly improve the success rate of paralinguistic task processing. 

In case of aggregation conbination, we can see that combining three techniques will always improve our results in any paralinguistic task. On the other hand the combination of four techniques will behave  inconsistently. Although it improves the results on the developement set, the results on the test set are often dicreasing. This suggests that the generalisation ability of our model is also decreasing. For this reason, when choosing the number of aggregations to combine, it is worth taking into account Occam's razor principle, which states that unnecessarily complex models should not be preferred against simpler ones.

Best regards, 
Mercedes Vetráb

Reviewer 2 Report

The authors present a hybrid method that combines automatic speech recognition and paralinguistic approaches, which is able to handle both of handling varying-length utterances with traditional classifiers and training models on relatively small corpora. Proposed feature extraction technique outperforms the widely-used x-vector method used as the baseline, independently of the actual paralinguistic task investigated. 

The topic is relevant in the field of computational paralinguistic. In result proposed technique from the preprocessing of the automatic speech recognition corpus to the classification of the paralinguistics corpus can be considered as a resource-efficient approach.

The authors trained a state-of-the-art hybrid acoustic HMM/DNN model on a large ASR corpus and then used the DNN part to extract frame-level embeddings from smaller paralinguistics corpora. The authors used statistics computational techniques to aggregate these features into utterance-level.

In my opinion, for further research it is better to use the existing paralinguistic corpus of the Hungarian language or create your own.

Conclusions are consistent with the evidence and arguments presented and correspond to the main question.

Obscure reference in introduction (Line 29. «estimating sleepiness [17]». Possible error in the bibliography 17 (17. Egas-López, J.V.; Gosztolya, G. Deep Neural Network Embeddings for the Estimation of the Degree of Sleepiness. In Proceedings 432of the Proceedings of ICASSP; , 2021; p. accepted.)

Error in text line 118, 254.  There are «spectograms». Need «spectrograms». I think it will be right. Maybe there is such an error in the text.

Author Response

Dear Reviewer,

First, we would like to thank you for your detailed review. We have corrected the citation error and typos you mentioned.

I would like to clarify our further research possibilities. We will add this point into our paper too. 

The possible limitations of our approach includes the potential language dependency of the extracted embeddings. For further research directions, we see several opportunities. Although our results were competitive even with a Hungarian HMM/DNN hybrid acoustic model for German tasks, and although the x-vector method (used as feature extractor) showed language-independency tendencies before~\cite{egaslopez2021dnnembeddings}, the effect of using an acoustic model trained on the same language should be studied in the future. On the other hand, each of the databases studied here is German-speaking, it could potentially be investigated whether aggregations computed from x-vector embeddings behave similarly on databases of different languages. Also, it is unclear how the amount of training material affects the quality of the extracted features. Furthermore, training a DNN is inherently a stochastic procedure due to random weight initialization; therefore, the variance in the classification performance might also prove to be an issue. We plan to investigate these factors in the near future. Another possible direction is whether these aggregations computed from different neural network embeddings follow a similar trend. 

Best regards, 
Mercedes Vetráb

Reviewer 3 Report

Attached

Need to be improved

Author Response

Dear Reviewer,

First, we would like to thank you for your detailed review. 

I would like to clarify the bellow mentioned points. We also modified the paper accordingly.

1-2 and 6-7-8 and 15. 
We clarified the introduction. We have made our workflow and model description more detailed. These are major changes in the firs part of the papaer. We reworked figure 1 and 2. 

3. We reorganised the introduction section and remove the figure reference.

4. In the field of paralinguistic, we need to associate different lengths of audio recordings with a single label. Here, very small training databases are typical, and therefore traditional classifier models (which can be well trained on very little data) are used instead of end-to-end neural networks. However, these traditional models can only process fix-length inputs, so we need to convert varying-lengths recordings into fix-length feature vectors. This transformation is called aggregation, so this name refers to a process rather than a specific mathematical method.  

5. We reworked figure 2 for support better understanding.

9-10. We are not working with speech enhancement. Paralinguistic refers to the non-verbal aspects of human communication such as tone of voice and other vocal cues. These cues play an essential role in understanding human communication and can significantly impact the meaning and interpretation of spoken language. Paralinguistic features are often extracted from audio data in machine learning applications, using techniques such as speech analysis and audio signal processing. 

11. "The manuscript is having too many sections and sub sections. The entire manuscript need
to restructured for better readiability."

We appreciate the observation, but we must disagree here. The manuscript has only sections and subsections, no third-level sub-subsections are used anywhere. In Section 1, the subsections belong to the different steps of the feature extraction process, being over 2 pages long overall, which should be broken into subsections in our opinion. Similary, Section 2 (one and a half page long) describes the databases used, where the subsections correspond to the databases. The experimental section (Section 3) also contains five topics -- here using subsections, in our opinion, improves the readability of the manuscript, as the reader cannot mix up, of example, embedding aggregation with the classification details.  The experimental results and the combined results are also broken up to subsections corresponding to each database.

12. UAR measures the average recall across all classes without considering class imbalance. To calculate UAR, you compute the recall for each class and then take the average across all classes. Recall, also known as sensitivity or true positive rate, is the proportion of true positive instances (correctly identified instances) out of all actual positive instances. UAR is called "unweighted" because it treats each class equally, regardless of class size or prevalence. This makes it suitable for datasets with imbalanced class distributions, where some classes may have significantly fewer instances than others. It provides a balanced view of the overall performance of a classification system, taking into account the performance across all classes equally.
In an emotion recognition task with an imbalanced dataset (Happy: 500, Sad: 300, Neutral: 2000), accuracy can be misleading. For instance, a classifier that predicts the majority class (Neutral) for all instances would have high accuracy (2000/2800 ≈ 71.4%). However, UAR (Unweighted Average Recall) gives a better evaluation by considering the recall for each class separately. In this case, UAR would indicate poor performance (UAR ≈ (0 + 0 + 1) / 3 ≈ 0.333) as the classifier fails to identify instances of the minority classes (Happy and Sad) while performing well on the majority class.

13. We appended a figure about the baseline x-vector model. Now it is represented as Figure 3.

14. We repalced the label description in Figure 4-5-6 for support better understanding.

16. In this field researchers tend to use task specific aggregations and only the most popular metrics such as mean and standard deviation are still being considered. Our purpose to show there are other efficient techniques which handles different paralinguistic subtopics.

17. Different aspects were taken into account when selecting the databases. On the one hand, we preferred databases that were easily accessible to the research community, and thus the databases used in the ComPare challenge were chosen. The other aspect was that they should be easily comparable, which is why we chose 3 German language databases, so  possible language differences would not affect the results.

18. We reffered Figure 1 in the end of the introduction section.
Based on our previous studies [ 28, 29], we developed a method showed in Figure 1.
that combines ASR and paralinguistic approaches. For frame-wise computing we followed 
standard ASR principles and we used DNNs to perform a frame-level feature extraction. 
Afterwards, to aggregate these features, we used more or less traditional computational 
paralinguistics techniques like standard-deviation and kurtosis. After, we employed SVM 
models to do the classification task.

19. The possible limitations of our approach includes the potential language dependency of the extracted embeddings. For further research directions, we see several opportunities. Although our results were competitive even with a Hungarian HMM/DNN hybrid acoustic model for German tasks, and although the x-vector method (used as feature extractor) showed language-independency tendencies before~\cite{egaslopez2021dnnembeddings}, the effect of using an acoustic model trained on the same language should be studied in the future. On the other hand, each of the databases studied here is German-speaking, it could potentially be investigated whether aggregations computed from x-vector embeddings behave similarly on databases of different languages. Also, it is unclear how the amount of training material affects the quality of the extracted features. Furthermore, training a DNN is inherently a stochastic procedure due to random weight initialization; therefore, the variance in the classification performance might also prove to be an issue. We plan to investigate these factors in the near future. Another possible direction is whether these aggregations computed from different neural network embeddings follow a similar trend. 

Best regards, 
Mercedes Vetráb

Round 2

Reviewer 3 Report

All comments are addressed